# VirB, a transcriptional activator of virulence in *Shigella flexneri*, uses CTP as a cofactor

Hammam Antar [1] & Stephan Gruber [1✉]

VirB is a transcriptional activator of virulence in the gram-negative bacterium *Shigella flexneri* encoded by the large invasion plasmid, pINV. It counteracts the transcriptional silencing by the nucleoid structuring protein, H-NS. Mutations in *virB* lead to loss of virulence. Studies suggested that VirB binds to specific DNA sequences, remodels the H-NS nucleoprotein complexes, and changes DNA supercoiling. VirB belongs to the superfamily of ParB proteins which are involved in plasmid and chromosome partitioning often as part of a ParABS system. Like ParB, VirB forms discrete foci in *Shigella flexneri* cells harbouring pINV. Our results reveal that purified preparations of VirB specifically bind the ribonucleotide CTP and slowly but detectably hydrolyse it with mild stimulation by the *virS* targeting sequences found on pINV. We show that formation of VirB foci in cells requires a *virS* site and CTP binding residues in VirB. Curiously, DNA stimulation of clamp closure appears efficient even without a *virS* sequence in vitro. Specificity for entrapment of *virS* DNA is however evident at elevated salt concentrations. These findings suggest that VirB acts as a CTP-dependent DNA clamp and indicate that the cellular microenvironment contributes to the accumulation of VirB specifically at *virS* sites.

[1] Department of Fundamental Microbiology (DMF), Faculty of Biology and Medicine (FBM), University of Lausanne, 1015 Lausanne, Switzerland.
✉email: stephan.gruber@unil.ch

*S*higella flexneri is the causative agent of the diarrheal disease shigellosis. It is a gram-negative bacterium that invades the epithelial lining of the intestinal tract[1]. *S. flexneri* contains a large 'invasion' plasmid pINV (~220 kb) encoding several virulence factors including components of a Type III secretion system and its effectors[2]. pINV encodes a major virulence factor, VirB, an unconventional transcription regulator[3]. At 37 °C, the physiological temperature of the host organism, VirB is expressed and activates the virulence programme of *S. flexneri*[4,5]. It is thought to counteract the transcriptional silencing of pINV mediated by the chromosomally expressed nucleoid associated protein, H-NS[6–10]. The molecular mechanisms underlying this de-repression are not entirely clear. Some models propose that VirB binds to specific DNA sequences and remodels the H-NS-DNA complexes making DNA more accessible for transcription (Fig. 1a)[11,12]. Moreover, a recent study suggests that VirB triggers a change of DNA supercoiling of plasmid DNA in vivo, which is dependent on the VirB DNA binding sites, here designated as *virS* sites, thus proposing a new mechanism by which VirB could be offsetting the H-NS-dependent transcription silencing[13]. Another recent study showed that GFP-tagged VirB proteins forms discrete fluorescent foci in the cell that are dependent on the presence of the large invasion plasmid[14]. VirB focus formation is believed to stem from the accumulation of VirB on the target sites found on pINV, including one in the *icsP* promoter (herein called *virS*^icsP). The *virS*^icsP site is sufficient for focus formation since a *Shigella* strain lacking pINV but carrying a plasmid with *virS*^icsP forms GFP-VirB foci. A crystal structure of the VirB middle domain (M domain) shows its helix-turn-helix motif bound to a DNA sequence found in the *icsB* promoter, another *virS* site (herein called *virS*^icsB)[12]. However, other studies have suggested that VirB might bind different sequences, so the mechanism of targeting remains poorly understood[9,10,15].

Based on domain organization and sequence similarity, VirB belongs to the superfamily of ParB proteins which are involved in DNA partitioning as part of ParABS systems[9,12,16] (Figs. 1b and 2a). In fact, pINV encodes for two ParB homologs, which share 39 % sequence identity with one another, and 41 and 58% identity with the *E. coli* P1 prophage ParB protein and somewhat lower similarity with chromosomally encoded ParB proteins (Supplementary Fig. S1a). The gene encoding for the more closely related homolog of ParB^P1 (58 % identity; herein referred to as ParB^pINV) is located downstream of a ParA homolog likely promoting

plasmid partitioning as a canonical plasmid ParABS system. The other one (41% identity with ParB^P1) is the transcription regulator VirB (Fig. 2a, Supplementary Fig. S1a).

ParABS systems comprise *parS* DNA sequences found near the origin of replication on the chromosome or plasmid, as well as the adapter CTPase ParB and the partitioning ATPase ParA[17,18]. ParA proteins form homodimers by binding ATP. ParA dimers associate with DNA in a sequence-unspecific manner. Multiple ParB dimers load onto DNA at the *parS* sites, together forming a ParB/DNA partition complex ('bacterial centromere'). This partition complex follows a ParA protein gradient on the bacterial chromosome to become equidistantly positioned within the cell. Once it interacts with ParA, it stimulates ParA ATP hydrolysis thus converting ParA dimers into monomers that dissociate from chromosomal DNA[19–21]. Following this diffusion ratchet mechanism, ParABS promotes chromosome partitioning and faithful DNA segregation[22].

In addition to being a DNA binding protein, ParB protein uses the unusual ribonucleotide cofactor CTP to mediate its functions in chromosome partitioning (Fig. 1b)[23–25]. ParB is an enzyme that binds and hydrolyses CTP using conserved motifs in the N-terminal domain ('N domain'). Two CTP molecules are sandwiched between two N domains of a given ParB dimer[23–25]. *parS* DNA binding greatly stimulates the formation of a N domain dimer interface by relieving ParB self-inhibition[26]. This N-domain engagement turns ParB dimers into closed clamps that topologically entrap DNA and are thus able to slide onto the *parS*-flanking DNA covering up to 15 kilobases large regions around *parS*. On the other hand, hydrolysis of CTP to CDP and inorganic phosphate destabilizes N-domain engagement allowing for ParB turnover thereby preventing the excessive spreading on DNA and recycling any DNA-free ParB clamps[26–28]. Moreover, ParB clamps have been suggested to recruit other ParB dimers, in a CTP-dependent manner, to load on DNA distal from *parS*[29]. Altogether, CTP-dependent loading and one-dimensional sliding of ParB as well as dimer-dimer recruitment is thought to allow for focus formation near the origin of replication (partition complex) to support faithful chromosome segregation.

VirB shares many features with ParB proteins (domain organization; accumulation in foci dependent on a cognate targeting site; see below for further similarities), but it lacks an obvious ParA-type partner protein and is not thought to contribute to DNA partitioning. A comparative analysis of VirB and ParB proteins may thus help to elucidate features that specifically

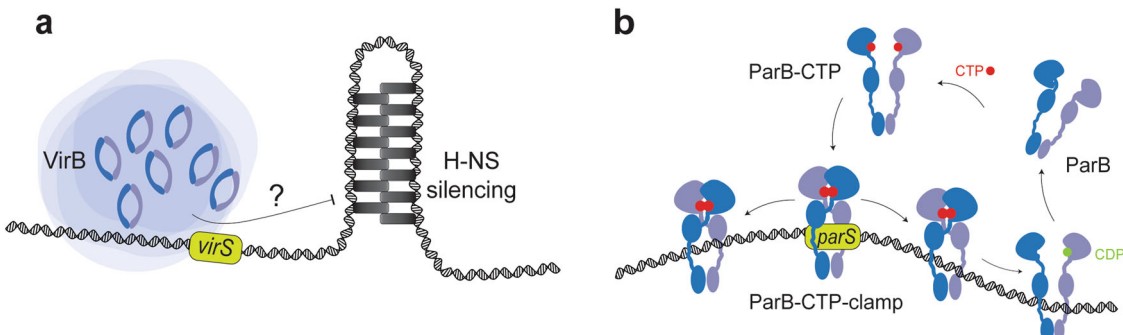

**Fig. 1 Schematic representation of VirB and ParB mode of action. a** Schematic representation of a model depicting the role of VirB in counteracting transcriptional silencing of pINV by H-NS. VirB is hypothesized to counteract the silencing effects of H-NS, a nucleoid structuring protein. The exact mechanisms remain unclear, but potential models suggest that VirB may enhance DNA accessibility for transcription by sterically blocking H-NS or changing DNA conformation or supercoiling (not indicated). **b** A schematic representation of ParB partition complex formation. In the absence of CTP binding, ParB exists in an open autoinhibited state. Upon binding to CTP and *parS* DNA, ParB self-inhibition is relieved resulting in the formation of ParB N domain dimer interface. This engagement of the N domain leads to the closure of ParB dimers, forming closed clamps that topologically entrap DNA and can slide onto *parS*-flanking DNA. Hydrolysis of CTP to CDP and inorganic phosphate destabilizes the N dimer engagement, promoting ParB release and turnover. This turnover mechanism prevents excessive spreading of ParB on DNA and allows for recycling of DNA-free ParB clamps.

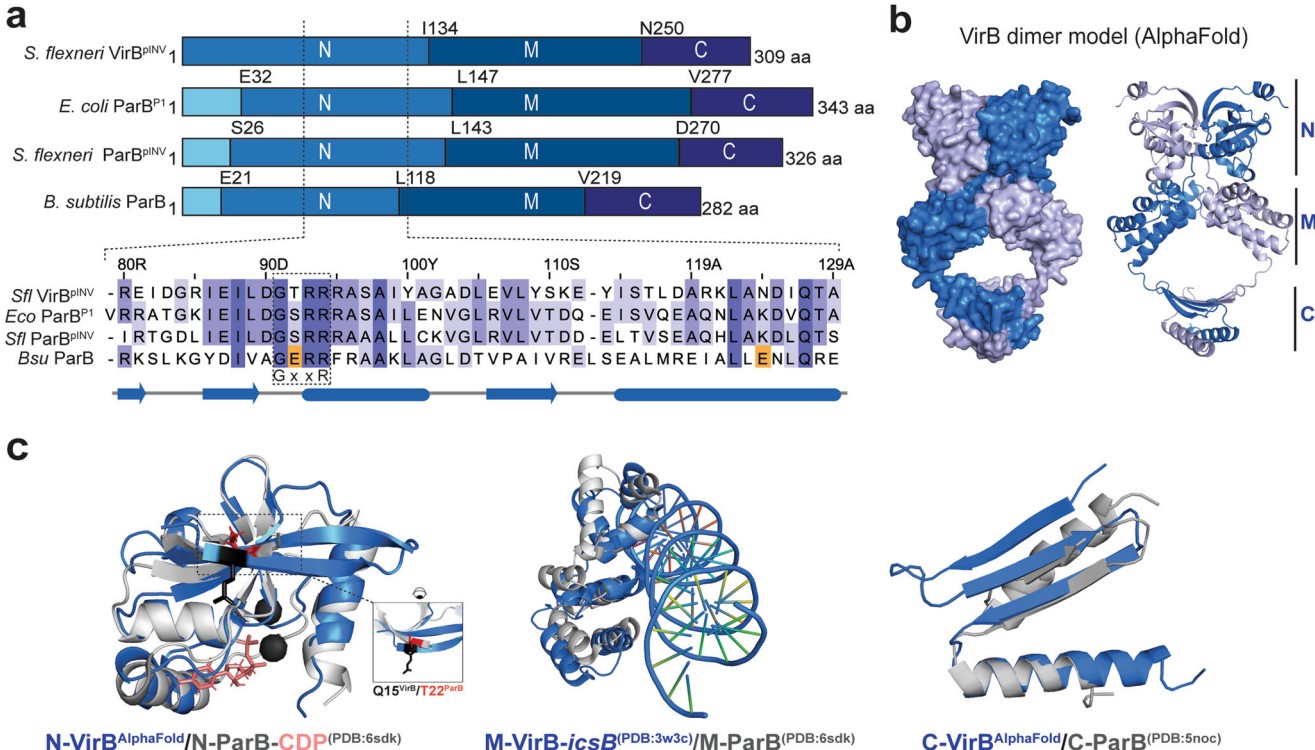

**Fig. 2 VirB shares sequence features and structural similarity with ParB. a** Domain organization of *Shigella flexneri (Sfl)* VirBpINV, *Escherichia coli (Ec)* ParBP1, *Sfl* ParBpINV, and *Bacillus subtilis (Bsu)* ParB (top panels). Sequence alignment of a part of the N domain of the four proteins including the conserved GxxR motif. The acidic residues responsible for CTP hydrolysis in *Bsu* ParB are highlighted in orange. **b** AlphaFold prediction of full-length *Sfl* VirB dimer displayed as surface and cartoon representation on the left and right panel, respectively. The prediction identifies three distinct domains (N, M, and C). **c** Separate superimposition of the three domains of *Sfl* VirB (blue) with the respective domain of *Bsu* ParB (gray). The N and C domains of VirB are AlphaFold predictions. The VirB M domain is from a crystal structure bound to the DNA targeting site, *virS*icsB (PDB:3W3C)[12]. The three domains of *Bsu* ParB are from available structural data (PDB:6SDK for N- and M, and PDB:5NOC for C domain).

support DNA partitioning or gene regulation. Here, we uncover strong structural and biochemical resemblance between VirB and ParB. We show that VirB specifically binds and hydrolyses CTP. Using site specific BMOE cross-linking, we show that CTP promotes VirB clamp closure in vitro, and that this closure is stimulated by specific and nonspecific DNA alike at physiological salt concentrations. We also show that VirB uses conserved residues in its CTP binding pocket to form foci in vivo (evident by imaging of a GFP-VirB fusion construction) that are dependent on the presence of a *virS* site in vivo. Loading of VirB specifically at a *virS* site was however only observed at artificially elevated salt concentrations in vitro. It is thus conceivable that site specific targeting of VirB (and maybe also ParB) in vivo is facilitated by one or more factors in the cellular microenvironment that remain to be explored.

## Results

### AlphaFold prediction of VirB shows strong resemblance to ParB protein structure. AlphaFold2 predictions of VirB dimers display a closed, clamp-like configuration that resembles the organization proposed for ParB dimers based on three separate ParB domains (N, M, and C domains) (Fig. 2b)[24,25,28,30]. We superimposed the VirB model with the crystal structure of the *Bacillus subtilis (Bsu)* ParB N domain bound to CDP (PDB: 6SDK). The superimposition showed close resemblance between the domains also revealing a pocket in VirB that may accommodate a ligand (Fig. 2c). This pocket is formed by sequences including the GxxR motif that is known to support CTP binding in ParB (Fig. 2a and Supplementary Fig. S1a). Published structures of ParB and VirB M domains (PDB: 6SDK and 3W3C,

respectively) as well as the C domains (PDB: 5NOC for ParB and an AlphaFold prediction for VirB) aligned well (Fig. 2c). We conclude that VirB shares multiple features with ParB proteins (including in the nucleotide-binding domain), implying that it could form partition complex-like nucleoprotein structures similar to ParB.

### VirB binds and hydrolyses CTP. We next addressed whether VirB binds any of the four ribonucleotides in vitro. Full-length VirB was recombinantly expressed and purified by N-terminal tagging with GFP. After removal of the GFP tag by proteolytic cleavage, full-length VirB was isolated. Supplementary Figure S1b displays the Size Exclusion Chromatography (SEC) profile of the purified protein showing a single elution peak. The peak fraction was collected and used for subsequent experiments. The purity of the protein was confirmed by SDS-PAGE analysis showing a single band corresponding to the molecular weight of VirB. Based on measurements with isothermal titration calorimetry, VirB bound CTP with relatively high affinity (Kd ~1 μM), while binding of the other ribonucleotides was not detected in this assay (Fig. 3a).

We then tested whether VirB can hydrolyze CTP (or one of the other three ribonucleotides) by measuring the release of free phosphate using endpoint colorimetric detection by Malachite green. The protein showed low but noticeable basal levels of CTP hydrolysis in the absence of DNA (with an estimated rate of ~1 CTP hydrolyzed/16 min per VirB monomer) (Fig. 3b). Upon addition of 40 bp DNA fragment containing *virS* sequences (*virS*icsp or *virS*icsB), CTP hydrolysis was poorly but significantly stimulated (~2-fold increase in the estimated hydrolysis rate: ~1

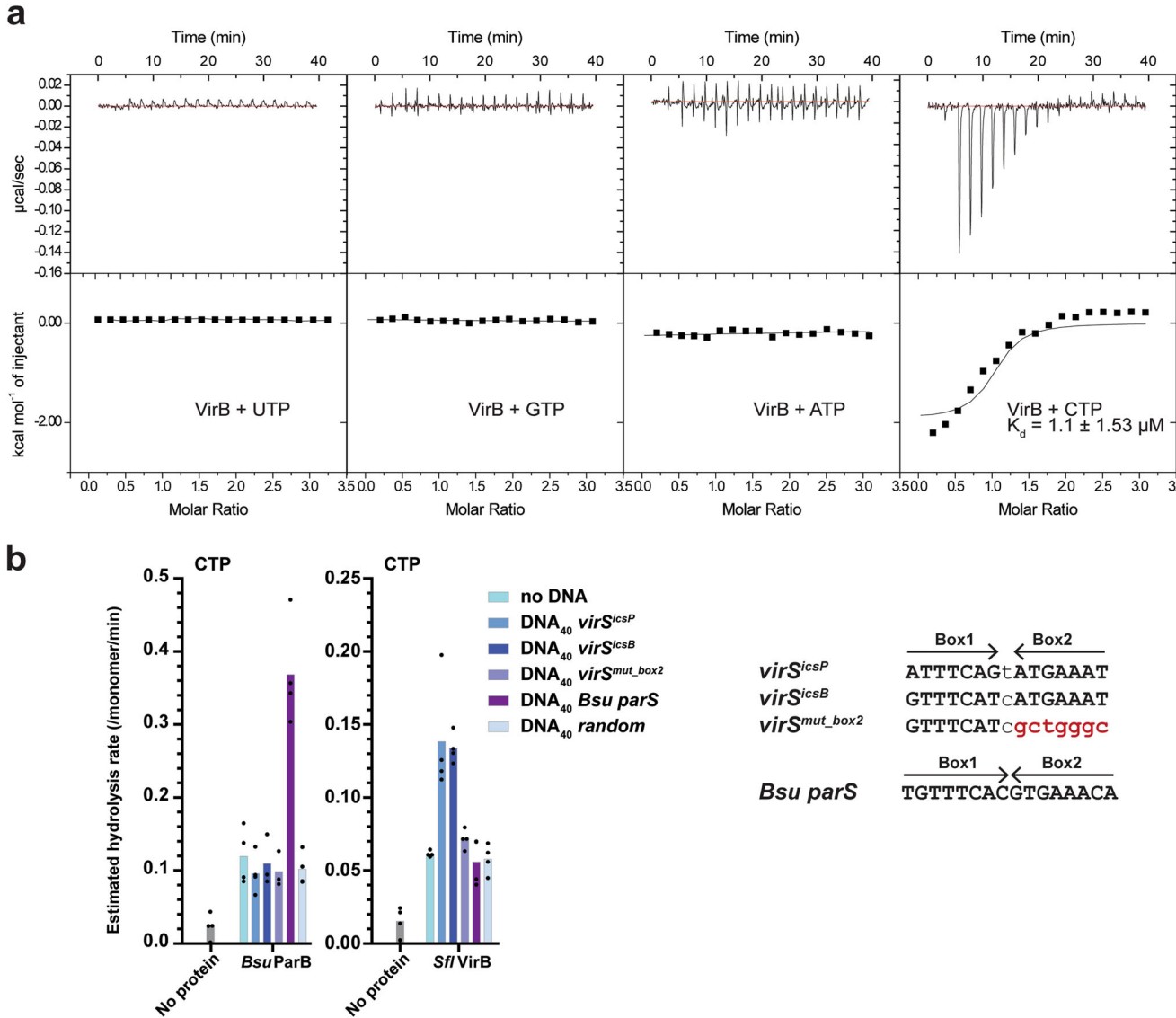

**Fig. 3 VirB binds and hydrolyses CTP. a** VirB-ribonucleotide affinity measurements by isothermal titration calorimetry (ITC). A typical titration curve is shown. The $K_d$ obtained from one experiment is provided. The interval indicates deviations of data points from the fit. **b** CTP hydrolysis rates by *Bsu* ParB (positive control) and *Sfl* VirB assayed by colorimetric detection of inorganic phosphate using Malachite Green Assay. Ten micromolar of protein was incubated with 1 mM CTP with or without 1 μM DNA$_{40}$. Mean values calculated from four repeat measurements are plotted. Individual data points are shown as dots. The right panel indicates the sequence of the various DNA sites tested.

CTP hydrolyzed/7 min). The stimulation was not detected when mutated or random DNA sequences were used (Fig. 3b). Usage of other ribonucleotide instead of CTP did not result in detectable levels of inorganic phosphate, suggesting that VirB specifically binds and hydrolyses CTP (Supplementary Fig. S2a). CTP hydrolysis is mildly stimulated by *virS* DNA sequences which we thus confirm to be specific recognition sequences for VirB protein (Fig. 3b).

**Efficient VirB clamp closure even in the absence of targeting DNA.** We hypothesized that VirB proteins, like ParB, form DNA sliding clamps that self-load onto the specific target sequences *virS*. To detect the engagement of the N domains, *i.e.*, closure of the VirB clamp, we employed site-specific cysteine cross-linking of purified VirB protein harboring a cysteine mutation. Based on superimposition of predicted VirB structures with published ParB structures (Fig. 2c), we chose VirB residue Q15 to be mutated to cysteine for BMOE cross-linking. Q15 falls

at the axis of symmetry of the N domain dimer and should thus support robust cross-linking in the closed form of VirB (Fig. 4a). C5 was removed by mutagenesis to eliminate any unwanted cross-reactivity. Purified VirB(C5S, Q15C) protein exhibited comparable CTPase activity, indicating that the mutations did not significantly hamper protein folding or stability (Supplementary Fig. S2a). In absence of ligands, a relatively small fraction of cross-linked VirB protein was detected (Fig. 4b) (~18%; lane 2). In the presence of CTP, more robust cross-linking was observed indicating that a significant fraction is found in a closed form (lane 3; ~40%) even in the absence of DNA. Addition of DNA further stimulated N domain engagement, particularly with *virS*$^{icsp}$, *virS*$^{icsB}$, and *virS*$^{mut\_box2}$ (lane 4–6; ~70%) but only slightly less so with other DNA sequences (lane 7 and 8; 55%). Of note, significantly lower levels of closed VirB clamps were detected in the presence of DNA when CTP was lacking (lanes 9–13; ~25–30%). The results indicate that cofactors are not strictly required for VirB N-domain engagement (at least as measured by Q15C

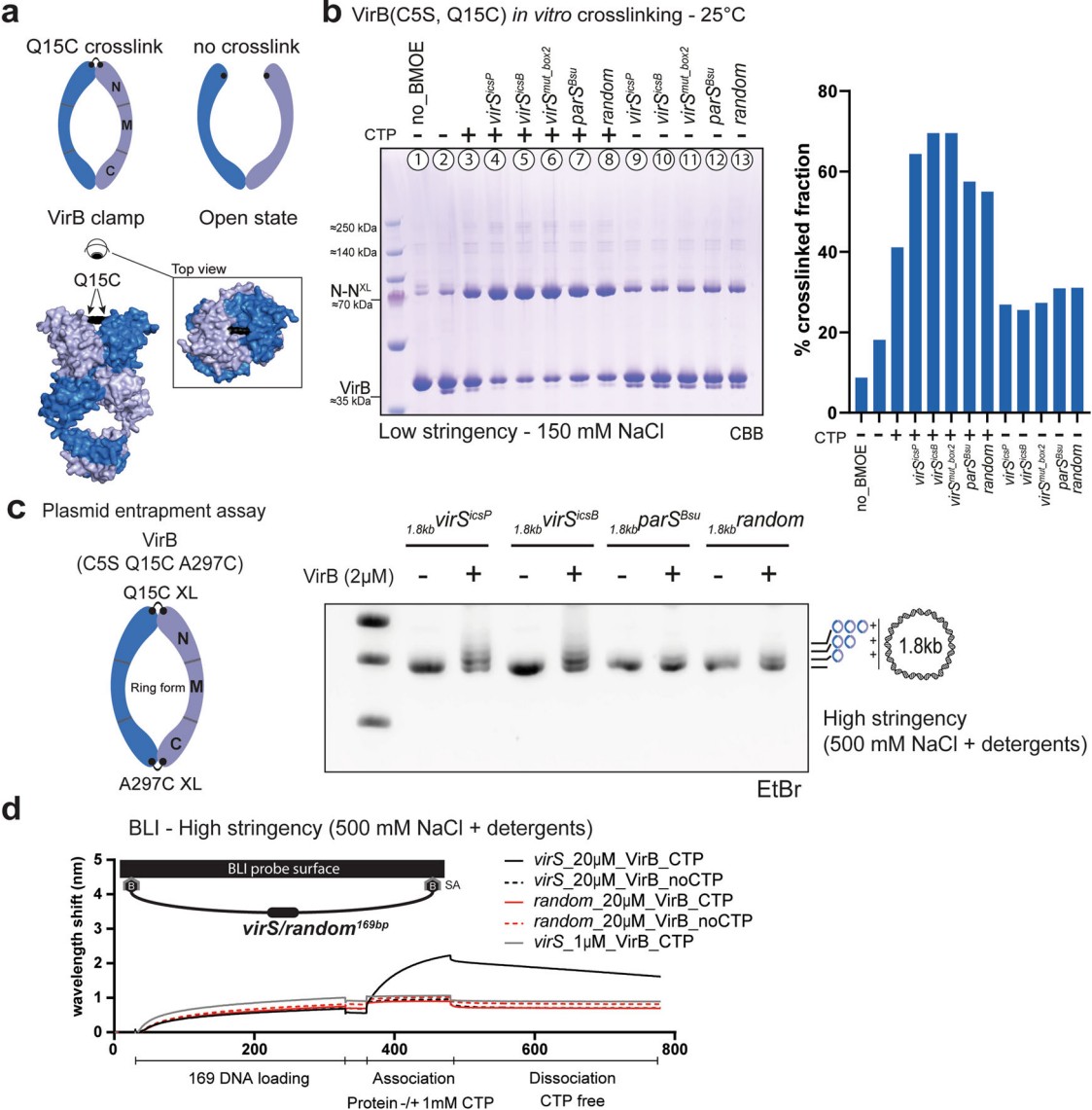

**Fig. 4 VirB N-gate closure and DNA loading. a** Model of full-length VirB dimer with the mutated Q15C residue highlighted in black. Endogenous cysteine residue (C5) residue was replaced by a serine residue (C5S) to avoid unspecific cross-linking. The upper panel offers a visual representation of the cross-linked (clamp) and non-cross-linked (open) states of VirB. **b** Gel analysis of cross-linking products of purified VirB(C5S, Q15C) at 25 °C. The different ligands tested are indicated. N-N^XL denotes cross-linked species of VirB. CBB, Coomassie Brilliant Blue. Quantification of cross-linked fractions is shown on the right panel. **c** Plasmid entrapment assay by VirB(C5S, Q15C, A297C) from BMOE-cross-linked DNA loading reactions. The buffer used here is high stringency buffer (25 mM HEPES/NaOH pH 7.5, 500 mM NaCl, 5 mM MgCl2, 1 mM DTT, 10 μM BSA, 0.01% [v/v] Tween 20)[31]. EtBr, Ethidium bromide. Plasmid species entrapping double cross-linked VirB dimers are marked. **d** Biolayer interferometry (BLI) analysis of VirB (1 and 20 μM) loading onto biotin-immobilized 169-bp *virS* or random DNA, measured in the presence of CTP (1 mM). The reaction buffer is the same as the one used in Fig. 4C (high stringency). The dissociation phase was carried out using the identical buffer as the association phase, but lacking VirB and CTP.

cross-linking), and that CTP alone can quite robustly support VirB clamp closure, unlike in canonical ParB proteins. Moreover, clamp closure is well stimulated by unspecific DNA (even at elevated salt concentrations without the addition of detergents (Supplementary Fig. S3a)). Other nucleotides failed to produce closed forms of VirB (Supplementary Fig. S3b). Of note, other cysteine residues (endogenous C5 or engineered C5S, I30C) resulted in comparable outcomes albeit with overall reduced cross-linking efficiency likely owing to their larger Cys-Cys distances. We performed the cross-linking experiment also at 37 °C (a temperature at which VirB is normally expressed in vivo) and obtained similar cross-linking efficiencies (Supplementary Fig. S3c). We conclude that CTP and DNA binding promotes N

domain engagement in VirB without displaying obvious preference for *virS* DNA.

**VirB specifically entraps *virS*-containing DNA under high stringency conditions.** We delved into VirB's capacity for topological DNA engagement. In addition to the cysteine residue at position Q15C (which cross-links the N domains of the VirB dimer), we introduced a cysteine residue in the C domain at position A297C (to cross-link the C domains of the VirB dimer). Cross-linking these two pairs of cysteine residues would result in a closed ring-like structure of a VirB dimer enabling to preserve topological DNA entrapment under protein-denaturing conditions. VirB (C5S, Q15C, A297C) was incubated with circular

plasmid DNA (1873 bp) in the presence of CTP and subsequently cross-linked with BMOE to capture topological DNA-protein interactions. We first performed these experiments at low stringency conditions (at a physiological salt concentration of 150 mM NaCl). The resulting electrophoresis gel analysis revealed significant amounts of covalently closed VirB rings associated with plasmid DNA with all four tested samples: containing *virS^{icsP}*, *virS^{icsB}*, *parS*, or random DNA (Supplementary Fig. S4a). Upon linearization of the plasmids using the NcoI restriction enzyme following BMOE cross-linking, the observed mobility shift of plasmid DNA was eliminated. This suggests a topological entrapment of DNA by VirB, as observed for ParB but lacking specificity for a DNA loading sequence (Supplementary Fig. S4b). However, when the experimental stringency was heightened (by increasing the salt concentration to 500 mM NaCl and addition of detergent, as previously reported[31]), the DNA mobility shift was more pronounced for plasmids containing a *virS^{icsP}* or *virS^{icsB}* site than in plasmids containing *parS* or random DNA sequences (Fig. 4c). To corroborate these findings, we employed biolayer interferometry (BLI) experiments in which VirB was loaded onto a double biotin-labeled 169-bp DNA fragment, which had been immobilized on a streptavidin-coated biosensor tip. Two distinct DNA fragments were engineered for this purpose, with one harboring a *virS^{icsP}* site and the other containing a random DNA sequence. At a physiological salt concentration (150 mM NaCl), VirB displayed loading on both DNA fragments with similar kinetics (Supplementary Fig. S4c). Notably, however, the dissociation of VirB from the *virS^{icsP}* containing DNA fragment (after loading at high VirB concentration) was significantly slower than from the control DNA indicating different modes of DNA association. Furthermore, when conducted under higher stringency conditions, BLI measurements showed specific loading of VirB onto DNA fragments containing the *virS* site (Fig. 4d)[31].

Altogether, we conclude from these experiments that CTP binding is required for robust VirB N-domain engagement (as measured by cysteine cross-linking) and that it is stimulated by DNA binding in a largely sequence-non-specific manner—in contrast to ParB—at least in our minimal reconstitution assay and under the reaction conditions used here. DNA loading and topological entrapment by VirB is specific to *virS* sites but only under high stringency buffer conditions tested here. The extent to which these conditions have relevance within the context of cellular physiology remains to be determined.

**GFP-VirB focus formation in vivo is CTP- and *virS*- dependent.** We finally investigated the subcellular localization of wild-type and mutant GFP-VirB fusion proteins using live-cell imaging. We utilized *E. coli* cells containing pLIBT7 (a low copy number plasmid) encoding N-terminal tagged GFP-VirB proteins. To probe the importance of the *virS* site, we generated two versions of the plasmid: one including a *virS^{icsP}* site and another devoid of it (Fig. 5a). In cells where the *virS^{icsP}* site was present, wild-type GFP-VirB cells (grown at 24 °C) exhibited punctuate fluorescence signals, being indicative of local protein clustering. Conversely, the absence of the *virS^{icsP}* site led to a pronouncedly diffuse fluorescence pattern, alluding to a more homogeneous distribution of the VirB protein throughout the cell. Furthermore, GFP-tagged versions of VirB bearing R93A and R94A mutations, residue mutations which have previously been implicated in compromising CTP binding by VirB[31,32], predominantly exhibited a uniform, diffuse fluorescence distribution, devoid of the discernible foci that characterized the wild-type protein (Fig. 5a). VirB(C5S, Q15C) mutant displayed focus formation comparable to wild type, demonstrating that this mutant is competent in protein cluster formation (presumably at *virS*). Collectively, these

findings suggest that VirB's ability to form foci in cells requires CTP binding and is highly specific for the cognate recognition sites on the plasmid.

## Discussion

In this paper, we show that VirB, a paralogue of ParB in *Shigella flexneri*, is a CTP-binding protein which also slowly hydrolyses CTP (Fig. 3). This marks transcriptional regulation as another example for the involvement of the unusual cofactor CTP originally found for ParB proteins and their roles in chromosome organization and partitioning. Yet another example is the nucleoid occlusion protein (Noc), a closely related paralog of chromosomal ParB in firmicutes, involved in the positioning of the cell division machinery in *Bacillus subtilis*, nucleates on the *parS*-like *nbs* sites using CTP[33]. It presumably does so by spreading on the DNA forming large nucleoproteins that hinder the assembly of the cell division machinery and direct it to the middle of the cell. In case of VirB, its local clustering at *virS* is likely needed to overcome the transcriptional silencing by H-NS. The ability to locally concentrate proteins by site-specific self-loading of DNA clamps—likely originating from the DNA partitioning ParABS systems—has thus been widely implemented in bacterial cells. The specific adjustments made in VirB for virulence regulation are however not well understood.

VirB binds with a relatively high affinity (low micromolar range) to CTP (when compared to ParB proteins) and hydrolyses it at a very low rate (about once every 5–10 min). The hydrolysis rate is mildly but specifically stimulated in presence of cognate DNA sites (*virS^{icsP}* and *virS^{icsB}*) in the assay described here. Similar levels of stimulation (albeit at slightly higher rates) have previously been observed with ParB proteins and *parS* sequences. In contrast, DNA stimulation of VirB N domain engagement is much less specific to *virS* DNA under physiological salt concentrations. A significant fraction of the protein is found in a closed state even in the absence of cognate DNA. This observation is puzzling because for ParB proteins, *parS*-stimulated clamp closure is believed to be a driving force for ParB accumulation near *parS* sites. ParB closure is thought to be the rate-limiting step of CTP hydrolysis due to the formation of a self-inhibited state by ParB. This self-inhibition is relieved by *parS* DNA binding by the latter acting as a catalyst for clamp closure[26,33]. This does not seem to be the case for VirB at least under the conditions tested here. VirB could be adopting an uninhibited state under our reaction conditions, making clamp closure efficient even in the absence of a catalyst. Consistent with this notion, we observed that VirB dimers entrap plasmid DNA in vitro regardless of the presence of *virS* sequences at salt concentrations mimicking physiological conditions (Supplementary Fig. S4a). Only when the experimental stringency was heightened by increase of salt concentration and addition of detergents did we observe a specificity of VirB to *virS* sites[31]. Likely other factors in the cellular microenvironment contribute for *virS*-specific targeting of VirB. We note that the essential ParB cofactor CTP has remained elusive for decades; similarly, other specificity cofactors of VirB might be missing in the reconstitution assays reported here.

It is generally thought that CTP binding and hydrolysis by ParB protein is crucial for *parS* DNA loading and the formation of partition complexes, and in turn for the functions in chromosome partitioning. Our data indicate that CTP binding (and maybe also CTP hydrolysis) contribute to the formation of VirB-*virS* nucleoprotein complexes (observed as foci by imaging) which in turn might be crucial for transcriptional regulation by VirB. Constructing mutants of VirB that are specifically defective in CTP hydrolysis would be insightful in addressing the specific function of this step in accumulating VirB at *virS* sites. CTP

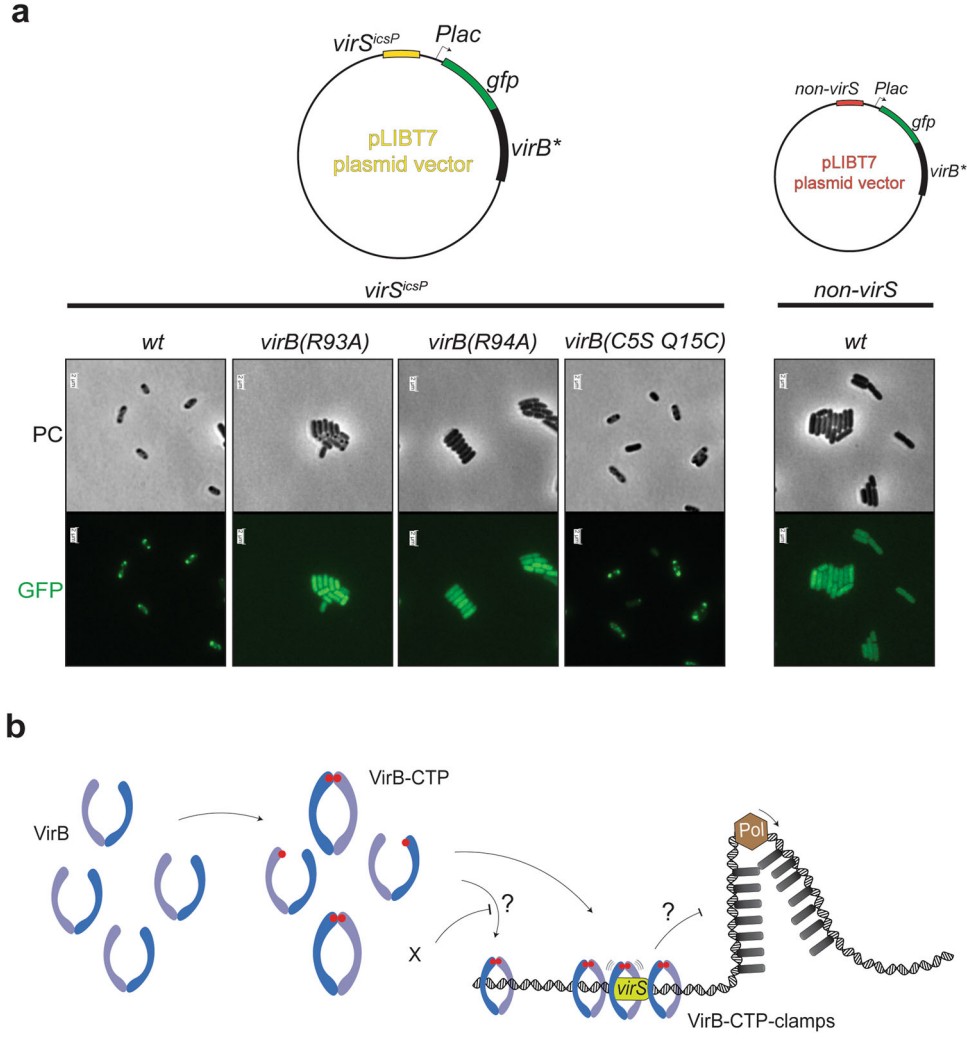

**Fig. 5 VirB focus formation in vivo and proposed mechanism. a** Example images of fluorescence microscopy of *E. coli* cells harboring pLIBT7 expression vectors encoding for different GFP-VirB proteins (in green) with or without *virS*icsP DNA site. The panel on the right displays the plasmid maps of the expression vectors used. **b** Proposed model for VirB CTP binding and clamp closure. Upon CTP binding, VirB engages in the N domain and accumulate at *virS* sites and spread to neighboring regions. X denotes putative factors in the cellular microenvironment that inhibits off-target accumulation of VirB. VirB focus formation at *virS* is hypothesized to counteract the silencing of virulence genes on pINV mediated by H-NS.

hydrolysis by ParB has been shown to promote bacterial centromere assembly by limiting 1) the accumulation of off-target ParB and 2) the excessive spreading of ParB on the DNA away from *parS*[26–28]. Curiously, the acidic residues contributing to CTP hydrolysis in chromosomal ParB proteins (GE$_{78}$RRY/F and E$_{111}$NLQR) are not found in VirB (or ParB$^{P1}$ and ParB$^{INV}$) (Fig. 2a and Supplementary Fig. S1a). This implies that the mechanism of CTP catalysis is not conserved and possibly has been adapted during evolution.

Altogether, we propose that VirB uses CTP as a cofactor, potentially together with so far elusive factors from the cellular microenvironment, to accumulate at high concentrations at the *virS* recognition sequences; the resulting VirB cluster is presumably needed to locally counteract the repressive function of H-NS proteins on the virulence plasmid for induction of virulence gene expression (Fig. 5b). Further work will be needed to elucidate the role of CTP usage by VirB on the virulence of *S. flexneri*.

### Materials and methods
**Expression and purification full-length proteins**. For *Bsu* ParB, expression constructs were prepared in pET-28 derived plasmids

by Golden-Gate cloning. Untagged recombinant ParB proteins were produced in *E. coli BL21-Gold (DE3)* grown in ZYM-5052 autoinduction media at 24 °C for 24 h. For *Sfl* VirB, the protein was cloned from *Shigella flexneri* (DSM4782) obtained from DSMZ (German Collection of Microorganisms and Cell Cultures). The cloning process involved the use of specific primers designed to amplify the *virB* gene (STN727 and STN728 and other primers to introduce mutations in *virB*. See Supplementary Table 1). Expression constructs were then assembled into pLIBT7 derived plasmids by Golden-gate cloning with an N-terminal GFP-tag[34]. GFP-tagged recombinant VirB proteins were produced in *E. coli BL21-Gold (DE3)* grown in TB-medium at 37 °C to an OD (600 nm) of 1.0 and the culture temperature was reduced to 24 °C. Expression was initiated with the addition of IPTG to a final concentration of 0.4 mM and was allowed to continue overnight, typically for 16 h. Purification of proteins was done as described before in[25,35]. In brief, cells were lysed by sonication in buffer A (1 mM EDTA pH 8, 500 mM NaCl, 50 mM Tris-HCl pH 7.5, 5 mM β-mercaptoethanol, 5 % (v/v) glycerol, and protease inhibitor cocktail (PIC, Sigma)). The first step for ParB purification involves adding ammonium sulfate to the supernatant until it reaches 40% (w/v) saturation and allowing it

to stir at 4 °C for 30 min. The sample is then centrifuged, and the supernatant is collected. Additional ammonium sulfate is added to the sample to reach 50% saturation, and it is allowed to stir at 4 °C for another 30 min. The pellet is collected by centrifugation and dissolved in buffer B (50 mM Tris-HCl pH 7.5, 1 mM EDTA pH 8 and 2 mM β-mercaptoethanol). For VirB, the first step involved running the supernatant on a homemade GFP-affinity column, and then the bound protein was proteolytically cleaved with an HRV-3C protease overnight. Both proteins were subjected to the same purification protocol thereafter. The sample was diluted with buffer B to a conductivity of 18 mS/cm and loaded onto a Heparin column (GE healthcare). The protein was eluted with a linear gradient of buffer B containing 1 M NaCl. Peak fractions were collected and diluted with buffer B to a conductivity of 18 mS/cm and loaded onto HiTrap SP columns (GE healthcare). A linear gradient of buffer B containing 1 M NaCl was used for elution. Peak fractions were collected and directly loaded onto a Superdex 200 16/600 pg column (GE healthcare) preequilibrated in 300 mM NaCl and 50 mM Tris-HCl pH 7.5. For cysteine mutants, 1 mM TCEP was added to the gel-filtration buffer.

**Isothermal titration calorimetry (ITC)**. ITC measurements were done using MicroCal iTC200 (GE Healthcare Life Sciences). The device was cooled to 4 °C before use. All measurements were performed in a buffer containing 150 mM NaCl, 50 mM Tris/HCl (pH 7.5), and 5 mM MgCl2. Purified protein peak fractions from the Superdex 200 16/600 pg column were collected, diluted 1:1 with buffer containing 50 mM Tris-HCl pH 7.5 and 10 mM MgCl2 to bring the final buffer to 150 mM NaCl, 50 mM Tris-HCl pH 7.5 and 5 mM MgCl2 and directly used for measurements. Both the measurement cell and the injection syringe were thoroughly cleaned with the same buffer. The measurement cell was filled with 280 μL of protein solution at a 30 μM monomer concentration, while the injection syringe was filled with buffer containing 0.5 mM NTP concentration or just buffer. The measurements were initiated after a 180-s delay, and the instrument settings were set to: reference power of 5 μcal/sec, stirring velocity of 1000 rpm, and "high feedback" mode. The raw data, expressed in kcal/mol, were presented as a Wiseman plot, and regression curves were calculated using a 1:1 nucleotide-to-protein monomer binding model when applicable. Origin software (GE Healthcare) was employed for fitting the measurement results using the equation:

$$\Delta Q(i) = Q(i) + \frac{dV_i}{dV_o}\left[\frac{Q_i + Q(i-1)}{2}\right] - Q(i-1)$$

In this equation, $V_i$ is the injection volume of ligand (nucleotides), $Vo$ denotes the cell volume, $Q(i)$ signifies the heat released from the $ith$ injection which is in turn calculated using the following equation:

$$Q = \frac{M_t \Delta H V_o}{2}\left[1 + \frac{X_t}{M_t}\frac{1}{KM_t} - \sqrt{\left(1 + \frac{X_t}{M_t} + \frac{1}{KM_t}\right)^2 - \frac{4X_t}{M_t}}\right]$$

Here, $K$ is the binding constant, $\Delta H$ is the molar heat of ligand binding, $Xt$ refers bulk concentration of nucleotide, and $Mt$ is the bulk concentration of ParB (moles/liter) in $Vo$. $K$ and $\Delta H$ were estimated by Origin, and $\Delta Q(i)$ for each injection was calculated and compared to the measured heat. To refine the estimates of $K$ and $\Delta H$, standard Marquardt methods were applied, and iterative adjustments were made until no further improvement in the fit could be achieved.

**Measurement of NTP hydrolysis by Malachite Green colorimetric detection**. NTP hydrolysis was measured as described in[26]. In brief mixtures of NTP (2x) with or without DNA40 (2x) and mixture of protein solutions (2x) were prepared in reaction buffer (150 mM NaCl, 50 mM Tris pH 7.5, 5 mM MgCl2) on ice. Equal volumes of each solution were mixed together (protein:ligand 1:1) using BenchSmart 96 (Rainin) dispenser robot and mixed through pipetting. Post-mixing, samples (containing 1 mM NTP, 1 μM DNA40, and 10 μM protein) were then incubated at 25 °C for 1 h; phosphate blanks were prepared in parallel. After incubation, samples were diluted four-fold by adding 60 μL of water, followed by mixing with 20 μL of working reagent (Sigma). The samples were then transferred to a flat-bottom 96-well plate. The plate was left to incubate for 30 min at 25 °C, after which the absorbance was measured at a wavelength of 620 nm. Absorbance values from the phosphate standard samples were used to plot an OD620 versus phosphate concentration standard curve. Raw values were converted to rate values using the standard curve, and absolute rates were determined by normalizing for protein concentration. Mean values and standard deviation were calculated from four replicates and presented as graphs on GraphPad Prism software.

**Preparation of 40-bp double stranded DNA**. To generate 40-bp double-stranded DNA, two complementary oligonucleotide strands at a concentration of 100 μM each were combined in a 1:1 ratio (STI706, STI707, STN815, STN816, SN817, STN818, STP408, STP409, STP531, STP532. See Supplementary Table 1). The resulting mixture was heated to 95 °C for 10 min and subsequently allowed to cool down to 25 °C.

**In vitro cysteine cross-linking**. A 2x CTP solution was prepared with or without DNA40 in reaction buffer (composed of 150/500/ 1000 mM NaCl, 50 mM Tris-HCl pH 7.5, and 5 mM MgCl2), and these mixtures were allowed to sit at room temperature for 5 min. A 2x protein solution (in the same buffer) was added to the mixture to obtain the following final concentrations: 10 μM protein, 1 mM CTP and 1 μM DNA40. The samples were then incubated for an additional 5 min at room temperature (unless stated otherwise) before adding 1 mM BMOE. After another 5 min at room temperature, the samples were quenched with β-mercaptoethanol (23 mM final). Loading dye was added and the samples were incubated at 70 °C for 5 min. Subsequently, they were loaded onto Bis-Tris 4–12% gradient gels (ThermoFisher). The bands were stained with Coomassie Brilliant Blue (CBB), and the relative band intensity was quantified by scanning and semi-automated analysis in ImageQuant (GE Healthcare).

**Plasmid DNA entrapment assay**. The plasmid entrapment assay was performed as described in[25] and similarly to in vitro cross-linking. The final concentrations of CTP, plasmid DNA or DNA40, and VirB protein were 250 μM, 50 nM, and 2 μM, respectively. Reaction mixtures were incubated at room temperature for 5 min and then treated with 1 mM BMOE for 5 min. After quenching with β-mercaptoethanol (23 mM final), the samples were divided into two halves. The first half was treated with 1x SDS loading dye for protein visualization and incubated at 70 °C for 5 min and loaded onto a WedgeWell Tris Glycine 4–12% gradient gel (ThermoFisher). Coomassie Brilliant Blue (CBB) staining was performed to detect proteins. The other half was mixed with 1x DNA loading dye and DNA detection involved running the samples onto 1% (w/v) TAE agarose gel containing ethidium bromide (0.5 μg/mL) and electrophoresed at 4 °C, 10 V/cm for 1–2 h. The agarose gel was then visualized using a Gel Doc XR+ (BioRad). In case of low stringency conditions,

the reactions were conducted in 150 mM NaCl, 50 mM Tris-HCl pH 7.5, and 5 mM MgCl2. In case of high stringency conditions, the reactions were conducted in 25 mM HEPES/NaOH pH 7.5, 500 mM NaCl, 5 mM MgCl2, 1 mM DTT, 10 µM BSA, and 0.01% [v/v] Tween 20 (as in ref. [31]).

**Cell culture and microscopy.** For microscopic localization of VirB protein, pLIBT7 expression constructs were transformed into *E. coli BL21-Gold (DE3)* cells. The vector plasmid encoded for N-terminally gfp-tagged *virB* and contained a *virS$^{icsP}$* DNA site (see Fig. 5a). Cells were grown in nutrient rich media (LB) to OD (600 nm) of 0.6 and expression of GFP-VirB was initiated with the addition of IPTG, at 24 °C, to a final concentration of 0.4 mM. Cells were then pelleted, washed two times with 1X PBS, and resuspended in 100 µL of fresh 1X PBS. Ten microliters of the cell suspension were then spotted on a glass slide coated with 1% agarose dissolved in 1X PBS. Slides were visualized using with a TIRF (Total Internal Reflection Fluorescence) microscope (Leica Microsystems).

**Biolayer interferometry (BLI).** Measurements were conducted in either low or high stringency buffer conditions (as detailed before) on BLItz machine (FortéBio Sartorius). We employed streptavidin-coated biosensors for all measurements, which were pre-hydrated in the reaction buffer for a duration of 10 min prior to loading. A baseline was first recorded by equilibrating the biosensor in 250 µL reaction buffer in a black 0.5 mL Eppendorf tube for 30 s. Subsequently, 4 µL of 100 nM biotin labeled double stranded *virS* or random DNA$_{169bp}$ were loaded on the biosensor for 5 min. 169 bp double-biotinylated DNA fragments were obtained by PCR amplification of *S. flexneri* genomic DNA with biotinylated primers STO396 and STO397 (See supplementary table 1), followed by gel purification. Following the DNA loading phase, the biosensor was washed once with the reaction buffer. Next, 2X VirB solution and 2x CTP solution were mixed 1:1 (yielding 1 mM CTP and the desired final concentration of VirB) and 4 µl of the mixture was loaded immediately on the biosensor for a duration of 2 min. The dissociation phase was then carried for 5 min in 250 µL protein-free and CTP-free reaction buffer. All data obtained were analyzed using the BLItZ analysis software and subsequently plotted on GraphPad Prism for presentation.

**Reporting summary.** Further information on research design is available in the Nature Portfolio Reporting Summary linked to this article.

## Data availability
The following publicly available datasets were used for protein superimposition: PDB accession nos. 6SDK, 5NOC, and 3W3C. Uncropped and unedited SDS page and agarose gel for Fig. 4 are shown in supplementary figure Fig. S5. Uncropped and unedited SDS page for Fig. S3 are shown in supplementary Fig. S6. Uncropped and unedited SDS page and agarose gel for Fig. S4 are shown in supplementary Fig. S7. Numerical source data for graphs are available via Mendeley https://doi.org/10.17632/d98vsk4k6p.1.

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

## Acknowledgements

We are grateful to members of the Gruber lab for stimulating discussions and comments on the manuscript and to Martin Thanbichler for sharing unpublished results. The authors acknowledge financial support from the Swiss National Science Foundation (197770 to S.G.).

## Author contributions

S.G. and H.A. conceived the project. S.G. acquired the funding. H.A. provided methodology and performed all experiments. S.G. and H.A. wrote the paper.

## Competing interests

The authors declare no competing interests.
