## [Peer Review File · Communications Biology]

Reviewers' comments:

Reviewer #1 (Remarks to the Author):

This is a short and sweet manuscript from Hammam Antar and Stephan Gruber. The two showed that VirB, a ParB-like protein, binds specifically CTP. VirB has a very slow but detectable CTPase rate. CTP and DNA (either virS or a non-specific sequence) can stimulate the clamp closure of VirB. At least in vitro, VirB could not distinguish well between a specific virS vs. a non-specific sequence. This final point is quite curious, that the authors proposed that there might be elusive factors that might prevent VirB clamping at off-site i.e. non-virS sequence in vivo. Overall, I think the manuscript reports a great finding that VirB binds CTP and there are implications of this co-factor in the ability of VirB to counter the anti-silencing effect from H-NS. All the experiments reported here were performed well and rigorously. There are a few places where the authors speculate/extrapolate too far without backed-up experimental evidence, I will point them out below for the authors and the editor to consider:

1) The sentence on "a so-far elusive factor that might prevent offsite DNA clamping in vivo" is perhaps too speculative, and without in vivo evidence. It is quite common that some purified DNA-binding proteins, outside the context of the cell, bind DNA very non-specifically but have specific binding sites without the help of any additional proteins/factors. The abundance of various other types of proteins and/or different cations or organic compounds etc... might simply shield off the non-specific DNA-binding activity of VirB in vivo and renders it binding to virS specifically. I would feel comfortable if the authors keep this sentence about an elusive factor and discuss it in the Discussion section, but perhaps it is more sensible to remove this speculative sentence from the Abstract and the Introduction?

2) Figure S3: plasmid entrapment assay. From the Materials & Methods, it seems that the experiment was done in low/normal salt concentration i.e. 100-150mM NaCl, is that correct? I wonder if this experiment can be repeated using a higher salt concentration (500 mM or 1M, similar to S2C) to see if entrapment will then be more specific to only the virS sequence? And an additional entrapment assay with a linearised plasmid will make the claim of VirB entrapping DNA even more convincing.

3) The sentence 161 "Our data suggest that CTP binding and hydrolysis also play a role in transcription regulation by VirB" should be modified because the authors have not made any connection experimentally with transcription activation in this work yet.

Minor comments:

- 1) Figure 2B: make the scale bars matched i.e. max value should all be 0.5?
- 2) There might be a case to split Figure 1 into 2 figures, and Figure 3 to 2 figures to improve readability
- 3) Several articles in the references have been published but are still reported as BioRxiv versions in this manuscript. It might be a good idea to go through the reference section and update them.

Reviewer #2 (Remarks to the Author):

Remarks to the Author (COMMSBIO-23-1635-review) :

In the manuscript, Dr Gruber and colleagues found that the virulence transcriptional regulator VirB in *Shigella* use CTP as a cofactor. This study used both experimental and bioinformatics methods to support the new discovery. However, I feel there is some caveats and supplements that will need to be fixed before acceptance of this publication. I hope my comments will be useful to the authors.

Major points:

1. In the introduction, it is known that relieve the inhibition of H-NS on the target genes is regulated by temperature, that is to say H-NS repress the expression of its target genes below 37°C and the inhibition is removed when the temperature reach 37°C. In this study, researchers found that CTP is the cofactor of VirB means that CTP is necessary for the function of VirB in Shigella. So the question is that if the specially bind of VirB and CTP is also regulated by temperature? Is the interaction specially happen when the temperature is 37°C for fulfill the regulatory function of VirB? In another word, is there any relevant between de-repression of H-NS and binding of CTP? Please supplement some data to support the opinion.
2. Although VirB belongs to the superfamily of ParB proteins, VirB is a transcription regulator and ParB is involved in DNA partitioning. What determines this different function, is the difference in sequence or in structure? What role does CTP play in this functional distinction? In other words, for both ParB and VirB can bind CTP and binding of CTP mediated the partitioning function of ParB, so is the binding of CTP give VirB the regulatory function or the interaction between CTP and VirB leads to the DNA binding domain on VirB more favorable to recognize and bind promoters? As shown in Fig 2, CTP hydrolysis rates by ParB is significantly higher than that by VirB, it indicate that the catalytic ability of VirB is lower than that of ParB, why? Is it the different change in structure after binding of CTP reduced the catalytic ability and promote the regulatory function of VirB? Please offer the relevant explanations and structural analysis.
3. Fig 1: The conserved GxxR motif is known to support CTP binding in ParB, will it affect the CTP binding after mutating the motif in VirB? If so, whether the regulatory function of VirB will be affected, such as counteract the inhibition of H-NS and activate its target genes expression?
4. Fig 3: Why the sites of Q15 and C5 were chosen for mutation? What are the important roles of the two sites in VirB, please explain.
5. As is known that VirB is necessary for the virulence of Shigella, so will it affect Shigella pathogenicity when VirB can not bind CTP (maybe after mutation of GXXR motif)? In other words, is the interaction between VirB and CTP is necessary for the virulence of Shigella? Because as shown in Fig 3B, the crosslink was still detected between VirB and virS although without CTP, it means that VirB can still partly activate the expression of the target genes. So I think it should clear and definite how much the disruption of binding CTP affects VirB function at transcription levels as well as Shigella virulence.

Minor points:

1. line 82: change "Bsu" to the full name of the bacteria for the first appearance.
2. Fig 1: change "Sfl", "Eco" and "Bsu" to the full name for the best or explain the full name in the figure legends.
3. Fig 2: What is "DNA40 scr" stand for?

Reviewer #3 (Remarks to the Author):

Antar and Gruber explored a potential structural mechanism for Shigella flexneri VirB-dependent

transcriptional regulation. Based on the similarities between VirB and a paralog plasmid partitioning protein (Bsu ParB), authors found that VirB also hydrolyzes CTP nucleotide while no hydrolysis was observed with other nucleotides. In addition, the authors showed that VirB-binding sequences (virS) stimulate CTP hydrolysis by VirB. VirB promotes transcription via mitigating H-NS repression; however, structural attributes of VirB anti-silencing remain unknown. To address this, the authors focused on the hypothesis of the formation of VirB clamps preceding DNA loading. The authors tested VirB dimerization by generating a VirB mutant (VirB-Q15C-C5S) that dimerizes via disulfide cross-linking. Authors demonstrated that CTP stimulates VirB clamp, yet it is not virS-specific. While CTP hydrolysis by VirB and CTP-stimulated VirB clamp formation are important findings, the current state of the manuscript does not address how VirB confers positive transcriptional regulation in *S. flexneri* and lacks physiological relevance. Below are points addressing these gaps:

- 1- A recent study discovered that VirB forms discrete foci in bacterial cells, a phenotype that relies on virS sequences. Authors should test foci formation by VirB-Q15C-C5S to show that this clamping VirB mutant can form fluorescent foci in *S. flexneri*.
- 2- As the authors indicated, the manuscript lacks VirB mutants that cannot hydrolyze CTP, which can be challenging to create in the short term. On the other hand, VirB-Q15C-C5S is a very useful tool. Another helpful tool would be to generate a VirB mutant that is not permissive to N-domain engagement or clamp closure. Did authors try to generate this mutant?
- 3- Authors should test the effect of other nucleotides on VirB-Q15C-C5S dimerization.
- 4- VirB is not as potent as Bsu ParB regarding CTP hydrolysis. Is it due to lacking E residue in the Bsu ParB GxxR motif? Authors should generate a VirB mutant harboring the E residue and assess related functions.

Reviewer #4 (Remarks to the Author):

The virulence factor VirB from *Shigella flexneri* was previously characterized for its key role in regulating transcription of several virulence factors encoded on the virulence plasmid pINV. Despite this, VirB is not a member of a canonical transcription factor family but rather is a member of the ParB family that functions in DNA partitioning. Previous data suggested that VirB functions in transcriptional regulation of virulence genes by countering the repressive function of the nucleoid protein H-NS. This paper builds off of previous biochemical and structural work to examine whether VirB has any of the biochemical properties of clamp loading proteins of the ParB family. Indeed, they found that, like ParB orthologs, VirB forms discrete foci in *S. flexneri*, binds and slowly hydrolyzes CTP, and stimulates clamp closure with DNA and CTP.

While this is an interesting finding, the authors did not address whether clamp closure is required for the in vivo function of VirB, specifically whether clamp closure is required to remodel HNS nucleoprotein complexes to counteract HNS silencing of pINV to control gene expression. Thus, the results presented here represent only an incremental advance.

Minor comment:

As written, it was not clear that the authors were truly measuring rates of NTP hydrolysis; it did not seem that they generated progress curves of Pi release over time, rather it appeared that they did fixed time point assays.

We would like to express our gratitude for the time and effort the reviewers dedicated to evaluating this manuscript. Their comments have been instrumental in refining and improving the manuscript. Here, we have addressed the concerns raised and hope that the revised version is stronger as a result and suitable for publication.

Note that we e also made additional minor edits to the manuscript text and figures.

Dear Dr Gruber,

Your manuscript entitled "VirB, a transcriptional activator of virulence in *Shigella flexneri*, uses CTP as a cofactor" has now been seen by 5 referees, whose comments are appended below. You will see from their comments copied below that while they find your work of potential interest, they have raised quite substantial concerns that must be addressed. In light of these comments, we cannot accept the manuscript for publication, but would be interested in considering a revised version that addresses these serious concerns.

We hope you will find the referees' comments useful as you decide how to proceed. Should further experimental data or analysis allow you to address these criticisms, we would be happy to look at a substantially revised manuscript. However, please bear in mind that we will be reluctant to approach the referees again in the absence of major revisions.

In particular, please note that the following revisions would be necessary for us to contact our referees again:

1) provide evidence that clamp formation is necessary *in vivo* (see co-report from R4 and R5) and

We provide evidence for the requirement of CTP binding for VirB focus formation *in vivo*. See details mentioned below. While this does not directly address clamp formation, we believe that (together with literature of ParB) this strongly points to a critical role for clamp formation in VirB.

2) provide more insight into the relationship/differences between ParB/VirB (see R2 and R3)

We have added corresponding statements to the introduction section: 'While VirB shares many features with ParB proteins (domain organization; accumulation in foci dependent on a cognate targeting site; see below for further similarities) it lacks a ParA-type partner protein and is not thought to contribute to DNA partitioning.' and further expanded on this in the discussion. Briefly, the data point to the conservation of all features need for local accumulation by DNA clamping. The only obviously lacking feature might be the N-terminal extension of ParB stimulating ParA-ATP hydrolysis (Figure 2A).

We are committed to providing a fair and constructive peer-review process. Do not hesitate to contact us if you wish to discuss the revision or if there are specific requests from the reviewers that you believe are technically impossible or unlikely to yield a meaningful outcome.

Referee expertise: Referee #1: Molecular biology/chromosome biology of bacteria

Referee #2: Molecular biology of pathogens

Referee #3: Biochemistry/molecular biology of bacteria (e.g. Shigella group)

Referee #4 and referee #5 (co-report marked R#4): Molecular genetics and biochemistry of gene regulation

Reviewers' comments:

Reviewer #1 (Remarks to the Author):

This is a short and sweet manuscript from Hammam Antar and Stephan Gruber. The two showed that VirB, a ParB-like protein, binds specifically CTP. VirB has a very slow but detectable CTPase rate. CTP and DNA (either virS or a non-specific sequence) can stimulate the clamp closure of VirB. At least in vitro, VirB could not distinguish well between a specific virS vs. a non-specific sequence. This final point is quite curious, that the authors proposed that there might an elusive factors that might prevent VirB clamping at off-site i.e. non-virS sequence in vivo. Overall, I think the manuscript reports a great finding that VirB binds CTP and there are implications of this co-factor in the ability of VirB to counter the anti-silencing effect from H-NS. All the experiments reported here were performed well and rigorously. There are a few places where the authors speculate/extrapolate too far without backed-up experimental evidence, I will point them out below for the authors and the editor to consider:

We thank the reviewer for the recognition of our work and the constructive criticism.

1) The sentence on “a so-far elusive factor that might prevent offsite DNA clamping in vivo” is perhaps too speculative, and without in vivo evidence. It is quite common that some purified DNA-binding proteins, outside the context of the cell, bind DNA very non-specifically but have specific binding sites without the help of any additional proteins/factors. The abundance of

various other types of proteins and/or different cations or organic compounds etc... might simply shield off the non-specific DNA-binding activity of VirB in vivo and renders it binding to *virS* specifically. I would feel comfortable if the authors keep this sentence about an elusive factor and discuss it in the Discussion section, but perhaps it is more sensible to remove this speculative sentence from the Abstract and the Introduction?

We agree that our statement could potentially be misleading. We therefore removed it from both Abstract and Introduction. We have instead phrased it as the following: "These findings suggest that VirB acts as a CTP-dependent DNA clamp and indicate that the cellular microenvironment contributes to the specific accumulation of VirB at *virS* sites."

2) Figure S3: plasmid entrapment assay. From the Materials & Methods, it seems that the experiment was done in low/normal salt concentration i.e. 100-150mM NaCl, is that correct? I wonder if this experiment can be repeated using a higher salt concentration (500 mM or 1M, similar to S2C) to see if entrapment will then be more specific to only the *virS* sequence? And an additional entrapment assay with a linearised plasmid will make the claim of VirB entrapping DNA even more convincing.

Many thanks for this suggestion. The revised version now contains the entrapment assay in high salt conditions + detergents (Fig. 4C). These buffer conditions are adapted from a recent preprint referred to as "high stringency buffer"[1]. In these conditions we indeed see a more pronounced (but still not exclusive) specificity of VirB to entrap *virS* containing plasmids. We also added performed DNA entrapment with linearization of DNA by *NcoI* restriction digest providing additional evidence for topological DNA entrapment (Fig. S4B).

3) The sentence 161 "Our data suggest that CTP binding and hydrolysis also play a role in transcription regulation by VirB" should be modified because the authors have not made any connection experimentally with transcription activation in this work yet.

We agree. We have changed the concerned sentence to the following: "Our data indicate that CTP binding (and maybe also CTP hydrolysis) contribute to the formation of VirB-*virS* nucleoprotein complexes (observed as foci by imaging) which in turn might be crucial for transcriptional regulation by VirB." We believe this is more fitting to the data we provide.

Minor comments:

1) Figure 2B: make the scale bars matched i.e. max value should all be 0.5?

We do agree that a direct side-by-side comparison with ParB would be preferable. However, changing the scale bar would make it harder to see the low yet detectable levels of CTP hydrolysis by VirB. We therefore prefer to maintain the current scale to support the readability of the data presented.

2) There might be a case to split Figure 1 into 2 figures, and Figure 3 to 2 figures to improve readability

These are great suggestions. We have split Figure 1 into Figure 1 and 2. We have split figure 3 into Figure 4 and 5. (Moreover, new panels have been added to Figure 4 and 5).

3) Several articles in the references have been published but are still reported as BioRxiv versions in this manuscript. It might be a good idea to go through the reference section and update them.

We have updated the refence list. Many thanks for the notification.

Reviewer #2 (Remarks to the Author):

Remarks to the Author (COMMSBIO-23-1635-review) :

In the manuscript, Dr Gruber and colleagues found that the virulence transcriptional regulator VirB in Shigella use CTP as a cofactor. This study used both experimental and bioinformatics methods to support the new discovery. However, I feel there is some caveats and supplements that will need to be fixed before acceptance of this publication. I hope my comments will be useful to the authors.

We thank the reviewer for the helpful comments and hope to have addressed all points sufficiently as explained below.

Major points:

1. In the introduction, it is known that relieve the inhibition of H-NS on the target genes is regulated by temperature, that is to say H-NS repress the expression of its target genes below 37°C and the inhibition is removed when the temperature reach 37°C. In this study, researchers found that CTP is the cofactor of VirB means that CTP is necessary for the function of VirB in Shigella. So the question is that if the specially bind of VirB and CTP is also regulated by temperature? Is the interaction specially happen when the temperature is 37°C for fulfill the regulatory function of VirB? In another word, is there any relevant between de-repression of H-NS and binding of CTP? Please supplement some data to support the opinion.

Thank you for raising this important and interesting point. VirB's expression is normally activated at 37°C. We have added this information in the introduction in the revised version of the

manuscript. Our experiments did not reveal any obvious temperature dependencies in VirB function/activity, neither in VirB cysteine cross-linking (see newly added figure S3C) or in VirB focus formation (here experiments are done at 24°C; previously reported findings at 28°C or 37°C). We thus propose that VirB expression itself rather than its activity is under temperature control.

2. Although VirB belongs to the superfamily of ParB proteins, VirB is a transcription regulator and ParB is involved in DNA partitioning. What determines this different function, is the difference in sequence or in structure? What role does CTP play in this functional distinction? In other words, for both ParB and VirB can bind CTP and binding of CTP mediated the partitioning function of ParB, so is the binding of CTP give VirB the regulatory function or the interaction between CTP and VirB leads to the DNA binding domain on VirB more favorable to recognize and bind promoters? As shown in Fig 2, CTP hydrolysis rates by ParB is significantly higher than that by VirB, it indicate that the catalytic ability of VirB is lower than that of ParB, why? Is it the different change in structure after binding of CTP reduced the catalytic ability and promote the regulatory function of VirB? Please offer the relevant explanations and structural analysis.

The differences in function between VirB, a transcription regulator, and ParB, which is involved in DNA partitioning, are not fully clear. Our manuscript focuses on the shared features (self-accumulation at targeting sites) rather than the specific functions that may be needed to either promote DNA partitioning or gene regulation. However, we do agree that those differences will be of high interest for future studies and our work will hopefully help to build a solid basis for revealing the distinctive features of VirB and ParB.

3. Fig 1: The conserved GxxR motif is known to support CTP binding in ParB, will it affect the CTP binding after mutating the motif in VirB? If so, whether the regulatory function of VirB will be affected, such as counteract the inhibition of H-NS and activate its target genes expression?

We have now shown that CTP binding mutants of VirB (R93A and R94A) fail to form foci and localize properly on the DNA like in the wild type (Fig. 5A). Although we don't test the effect of CTP binding on transcription activation mediated by VirB, other recent reports have suggested that there likely is a causal connection [1, 2].

4. Fig 3: Why the sites of Q15 and C5 were chosen for mutation? What are the important roles of the two sites in VirB, please explain.

The engineering of cysteine residues is indeed critical for measuring protein conformation by cysteine cross-linking. Our choice is largely based on structure predictions. *Bacillus subtilis* ParB residue T22C has been utilized for BMOE crosslinking experiments to detect closed form species of ParB [3, 4]. AlphaFold predictions of VirB indicate that Q15 is located on the symmetry axis of the N domain dimer in VirB (analogous to T22C on ParB). This suggests that it would be particularly effective for robust cross-linking when VirB is in its closed form. Thus, we specifically mutated Q15 to cysteine to facilitate BMOE-crosslinking. To avoid undesired cys-cys BMOE crosslinking, we changed the native cysteine at position C5 to serine (C5S), ensuring specific BMOE crosslinking only at the Q15C position. We have now indicated the position of these residues in Figure 2C.

5. As is known that VirB is necessary for the virulence of *Shigella*, so will it affect *Shigella* pathogenicity when VirB can not bind CTP (maybe after mutation of GXXR motif)? In other words, is the interaction between VirB and CTP is necessary for the virulence of *Shigella*? Because as shown in Fig 3B, the crosslink was still detected between VirB and virS although without CTP, it means that VirB can still partly activate the expression of the target genes. So I think it should clear and definite how much the disruption of binding CTP affects VirB function at transcription levels as well as *Shigella* virulence.

We acknowledge the importance of answering these questions. Unfortunately, these specific experiments fall beyond the scope of our team's research and hence beyond the tools and techniques we have currently available. We believe that teams with more specialized expertise will (and already have started to) address these questions.

Minor points:

1. line 82: change "Bsu" to the full name of the bacteria for the first appearance.

We changed it to the full name (*Bacillus subtilis*).

2. Fig 1: change "Sfl", "Eco" and "Bsu" to the full name for the best or explain the full name in the figure legends.

We changed them to the full name.

3. Fig 2: What is "DNA40 scr" stand for?

We changed it the terminology 'scr' (scrambled) to 'random'.

Reviewer #3 (Remarks to the Author):

Antar and Gruber explored a potential structural mechanism for *Shigella flexneri* VirB-dependent transcriptional regulation. Based on the similarities between VirB and a paralog plasmid partitioning protein (Bsu ParB), authors found that VirB also hydrolyzes CTP nucleotide while no hydrolysis was observed with other nucleotides. In addition, the authors showed that VirB-binding sequences (virS) stimulate CTP hydrolysis by VirB. VirB promotes transcription via mitigating H-NS repression; however, structural attributes of VirB anti-silencing remain unknown. To address this, the authors focused on the hypothesis of the formation of VirB clamps preceding DNA loading. The authors tested VirB dimerization by generating a VirB mutant (VirB-Q15C-C5S) that dimerizes via disulfide cross-linking. Authors demonstrated that CTP stimulates VirB clamp, yet it is not virS-specific. While CTP hydrolysis by VirB and CTP-stimulated VirB clamp formation are important findings, the current state of the manuscript does not address how VirB confers positive transcriptional regulation in *S. flexneri* and lacks physiological relevance. Below are points addressing these gaps:

We agree with this reviewer that we currently cannot explain how VirB confers transcriptional regulation in *S. flexneri*. However, we show more clearly in the revised version of the manuscript that CTP binding is critical for VirB function (for local accumulation of VirB on the virulence regulon underscoring its physiological relevance). Follow-up studies will hopefully reveal how local accumulation leads to anti-silencing.

1- A recent study discovered that VirB forms discrete foci in bacterial cells, a phenotype that relies on virS sequences. Authors should test foci formation by VirB-Q15C-C5S to show that this clamping VirB mutant can form fluorescent foci in *S. flexneri*.

We have tested this and is now showed in Figure 5A. The results confirm that C5S Q15C VirB reporter protein can form foci normally *in vivo*.

2- As the authors indicated, the manuscript lacks VirB mutants that cannot hydrolyze CTP, which can be challenging to create in the short term. On the other hand, VirB-Q15C-C5S is a very useful tool. Another helpful tool would be to generate a VirB mutant that is not permissive to N-domain engagement or clamp closure. Did authors try to generate this mutant?

Mutants that are specifically defective in N-domain engagement or clamp closure are not available for VirB (or ParB for that matter). As a proxy, we have looked at mutants that are defective in CTP binding (and in turn unable to form closed clamps at least in ParB proteins). The findings (Figure 5A) are consistent with the idea that N-domain engagement is required for focus formation (but other interpretations might likewise explain the data).

3- Authors should test the effect of other nucleotides on VirB-Q15C-C5S dimerization.

We have tested this and is now reported in supplementary figure S3B. the results show no significant stimulation of other nucleotides on the dimerization of VirB(C5S Q15C). The increase of stimulation in the presence of *virS* DNA is the result of DNA alone as reported in Fig. 4A where the DNA alone can stimulate VirB dimerization.

4- VirB is not as potent as Bsu ParB regarding CTP hydrolysis. Is it due to lacking E residue in the Bsu ParB GxxR motif? Authors should generate a VirB mutant harboring the E residue and assess related functions.

This is an intriguing idea. While we have not been able to create such a mutant in the course of the revision of the ms, we look forward to determining its activity and phenotype in future work.

Reviewer #4 (Remarks to the Author):

The virulence factor VirB from *Shigella flexneri* was previously characterized for its key role in regulating transcription of several virulence factors encoded on the virulence plasmid pINV. Despite this, VirB is not a member of a canonical transcription factor family but rather is a member of the ParB family that functions in DNA partitioning. Previous data suggested that VirB functions in transcriptional regulation of virulence genes by countering the repressive function of the nucleoid protein H-NS. This paper builds off of previous biochemical and structural work to examine whether VirB has any of the biochemical properties of clamp loading proteins of the ParB family. Indeed, they found that, like ParB orthologs, VirB forms discrete foci in *S. flexneri*, binds and slowly hydrolyzes CTP, and stimulates clamp closure with DNA and CTP.

While this is an interesting finding, the authors did not address whether clamp closure is required for the *in vivo* function of VirB, specifically whether clamp closure is required to remodel HNS nucleoprotein complexes to counteract HN-S silencing of pINV to control gene expression. Thus, the results presented here represent only an incremental advance.

We have now strengthened the manuscript by providing additional data that show the importance of CTP and *virS* DNA binding by VirB to form foci *in vivo*. The relationship between clamp closure and offsetting HN-S silencing is a compelling and exciting line of inquiry and we fully recognize the importance and significance of such experiments. However, we believe that this could be a topic of future specialized studies ideally pursued by research teams with greater expertise in the field of transcription regulation and *Shigella* virulence. We hope our current findings serve as a foundational step for those investigations.

Minor comment:

As written, it was not clear that the authors were truly measuring rates of NTP hydrolysis; it did not seem that they generated progress curves of Pi release over time, rather it appeared that they did fixed time point assays.

We indeed utilized fixed time point hydrolysis assay in our study. We state this more clearly in the manuscript and replaced 'rate' by 'estimated rate'. The primary rationale behind this choice was equipment and technical limitations. Of note, levels of inorganic phosphate release were measured 1hr after incubation at 25°C. Considering the low rates of hydrolysis, substrate depletion is not a valid concern, however, protein instability during the incubation period may lead to underestimation of the hydrolysis rates.

1. Jakob, S., et al., *The virulence regulator VirB from *Shigella flexneri* uses a CTP-dependent switch mechanism to activate gene expression*. bioRxiv, 2023: p. 2023.06.01.543266.
2. Gerson, T.M., et al., *VirB, a key transcriptional regulator of *Shigella* virulence, requires a CTP ligand for its regulatory activities*. mBio. **0**(0): p. e01519-23.
3. Antar, H., et al., *Relief of ParB autoinhibition by parS DNA catalysis and recycling of ParB by CTP hydrolysis promote bacterial centromere assembly*. Science advances, 2021. **7**(41): p. eabj2854.
4. Soh, Y.M., et al., *Self-organization of parS centromeres by the ParB CTP hydrolase*. Science, 2019.

REVIEWERS' COMMENTS:

Reviewer #1 (Remarks to the Author):

The authors have addressed all of my concerns. I have no further comment. Thank you.

Reviewer #2 (Remarks to the Author):

I am very grateful to see the concerns have been addressed. The author give an appropriate response to each question.

Reviewer #3 (Remarks to the Author):

This is a really interesting piece of work.

The authors addressed our comments.

We are looking forward to seeing this published soon.